# PARP1-Inhibition Sensitizes Cervical Cancer Cell Lines for Chemoradiation and Thermoradiation

**DOI:** 10.3390/cancers13092092

**Published:** 2021-04-26

**Authors:** Marloes IJff, Gregor G. W. van Bochove, Denise Whitton, Roy Winiarczyk, Celina Honhoff, Hans Rodermond, Johannes Crezee, Lukas J. A. Stalpers, Nicolaas A. P. Franken, Arlene L. Oei

**Affiliations:** 1Laboratory for Experimental Oncology and Radiobiology (LEXOR), Center for Experimental Molecular Medicine, Amsterdam University Medical Centers, P.O. Box 22700, 1100 DE Amsterdam, The Netherlands; m.ijff@amsterdamumc.nl (M.I.); g.g.vanbochove@amsterdamumc.nl (G.G.W.v.B.); denisewhitton@hotmail.nl (D.W.); roy_winiarczyk@hotmail.com (R.W.); celinahonhoff@hotmail.com (C.H.); h.rodermond@amsterdamumc.nl (H.R.); l.stalpers@amsterdamumc.nl (L.J.A.S.); n.a.franken@amsterdamumc.nl (N.A.P.F.); 2Department of Radiation Oncology, Amsterdam University Medical Centers, P.O. Box 22700, 1100 DE Amsterdam, The Netherlands; h.crezee@amsterdamumc.nl

**Keywords:** cervical cancer, LACC, DSB repair, PARP1-inhibition, cisplatin, radiosensitization, hyperthermia

## Abstract

**Simple Summary:**

Five-year survival rates from patients with locally advanced cervical cancer (LACC) are between 40% and 60%. These patients are usually treated with chemoradiation or radiotherapy in combination with hyperthermia (thermoradiation). The aim of our study was to enhance chemoradiation or thermoradiation by adding PARP1-inhibition to these conventional therapies. To study this, different cervical cancer cell lines were used to measure cell reproductive death and analyze DNA double strand breaks and cell death. By looking into the surviving fractions and DNA double strand breaks, our results suggest that PARP1-*i* sensitizes cervical cancer cells for the conventional therapies. The results of the live cell imaging suggest that effects are solely additive.

**Abstract:**

Radiotherapy plus cisplatin (chemoradiation) is standard treatment for women with locoregionally advanced cervical cancer. Both radiotherapy and cisplatin induce DNA single and double-strand breaks (SSBs and DSBs). These double-strand breaks can be repaired via two major DNA repair pathways: Classical Non-Homologous End-Joining (cNHEJ) and Homologous Recombination. Besides inducing DNA breaks, cisplatin also disrupts the cNHEJ pathway. Patients contra-indicated for cisplatin are treated with radiotherapy plus hyperthermia (thermoradiation). Hyperthermia inhibits the HR pathway. The aim of our study is to enhance chemoradiation or thermoradiation by adding PARP1-inhibition, which disrupts both the SSB repair and the Alternative NHEJ DSB repair pathway. This was studied in cervical cancer cell lines (SiHa, HeLa, C33A and CaSki) treated with hyperthermia (42 °C) ± ionizing radiation (2–6 Gy) ± cisplatin (0.3–0.5 µM) ± PARP1-inhibitor (olaparib, 4.0–5.0 µM). Clonogenic assays were performed to measure cell reproductive death. DSBs were analyzed by γ-H2AX staining and cell death by live cell imaging. Both chemoradiation and thermoradiation resulted in lower survival fractions and increased unrepaired DSBs when combined with a PARP1-inhibitor. A quadruple modality, including ionizing radiation, hyperthermia, cisplatin and PARP1-*i,* was not more effective than either triple modality. However, both chemoradiation and thermoradiation benefit significantly from additional treatment with PARP1-*i*.

## 1. Introduction

Cervical cancer is the fourth most common malignant disease in women worldwide [1,2]. Especially in low-income countries, many women suffer from cervical cancer because of insufficient infrastructure, lack of screening and a lack of trained personnel [3]. Approximately 85% of worldwide deaths from cervical cancer are found in the underdeveloped countries [2,4]. Cervical cancer is staged by the FIGO system. Stages IA1 up until IIA are considered to be the lower stages and are usually treated by surgery only. Stages IIB up until IVA are known as the locally advanced cervical cancer (LACC) stages. Most patients with LACC are treated with radiotherapy in combination with chemotherapy (chemoradiation), mostly with cisplatin (cDDP). However, five-year survival rates of patients with LACC remain poor, between 40% to 60%, despite modern combined modality and image-guided radiotherapy [5]. There is a group of patients in whom cDDP is contraindicated, for instance in women with renal failure, or in elderly and fragile patients [6]. These patients can be treated with radiotherapy in combination with hyperthermia (thermoradiation). During hyperthermia, the tumor temperature is increased to 41–43 °C for approximately 1 h. Thermoradiation yields similar results with respect to survival as chemoradiation [7,8,9,10]. These two LACC treatment regimens yield good clinical results, but problems still arise, such as systemic relapses, indicating the need to improve standard treatment [1].

Radiotherapy is used for approximately 50% of all types of cancer in combination with other therapies [11]. Radiotherapy causes DNA damage that eventually leads to cancer cell death, either through direct induction of DNA double-strand breaks (DSBs) or indirectly by the formation of free radicals [11,12,13]. Incorrectly repaired DNA damage can lead to mutations or cell reproductive death. There are different pathways involved in the repair of DNA DSBs. The question of which pathway is activated depends on the phase of the cell cycle, the complexity of the breaks and chromatin conformation [13]. There are two main DNA DSB repair pathways, homologous recombination and non-homologous end-joining (NHEJ) [14]. HR is considered to be an error-free pathway, requiring a sister chromatid as a template to repair the DSB. Therefore, HR is only active during the S- and G2-phases of the cell cycle, when a sister chromatid is available [14,15]. The NHEJ pathway can be subdivided into the classical NHEJ (cNHEJ) and the alternative NHEJ (AltNHEJ) pathway [13,16,17]. cNHEJ is active throughout the entire cell cycle since it does not rely on a template, but is prone to errors [18]. Whether the cell uses cNHEJ or HR to repair the DNA DSB during the S and G2-phase is largely dependent on DNA-end resection and is controlled by cyclin-dependent kinases (CDKs) [14]. AltNHEJ is only used when cNHEJ is not available, which typically occurs due to a lack of its key protein components [19]. Tumor cells will survive when the induced DSBs are repaired correctly. Combining radiotherapy with tumor targeting agents that interfere with the DNA repair pathways increase the number of residual DNA breaks and thus cause more cell death [20,21].

Cisplatin is a commonly used drug in a large variety of cancers. However, particularly in combination with radiotherapy, cDDP is associated with severe side-effects, which limit the further increase of drug or radiation dose [22]. High dosages of cDDP induce oxidative stress, which may lead to hepatotoxicity. cDDP also leads to cardiotoxicity such as degeneration and necrosis of cardiac muscle fiber cells with fibrous tissue reaction. Kidney tissue accumulates cDDP to a great degree, leading to nephrotoxicity [23]. Since the early 1980s, hyperthermia (HT) is used as a treatment in combination with radiotherapy or chemotherapy [24,25]. Mild HT is considered to be a local heating of the tumor up to 43 °C for approximately 1 h. Research has shown that HT increases radiosensitivity [26]. Cells in the S-phase, which are resistant to radiotherapy, are sensitive to HT [27], and cells in hypoxic areas are less sensitive to ionizing radiation, but were also shown to be more sensitive to HT [25]. Moreover, HT has been shown to inhibit BRCA2, one of the key players of the HR pathway [28,29]. This might be one of the reasons why HT enhances radiotherapy in clinical treatment [28,29]. Local or regional HT combined with radiotherapy improves tumor control and survival of women with cervical cancer, depending on the size of the tumor [10,24]. Besides, both HT and radiotherapy can be locally applied, indicating that the effects on healthy tissue are less severe. HT has fewer side effects compared to cDDP, which is why it is considered to be a good alternative in combination with radiotherapy. HT-related toxicity includes second and third-degree skin and subcutaneous burns, but these are usually self-limited [27].

PARP-inhibitors (PARP-*i*), e.g., olaparib, are therapeutic agents targeting poly(ADP-ribose) polymerase, which also gained interest in the field of cervical cancer therapy. One of the key enzymes is Polymerase 1 (PARP1), which is involved in AltNHEJ. PARP1 is also involved in single-strand breaks (SSBs) repair [30]. Olaparib is a clinically used PARP1-*i* chemotherapeutic agent that is especially effective for treatments of BRCA1 or BRCA2 negative cancers, such as ovarian and breast cancer types [30,31,32]. PARP1-*i* alone or in combination with platinum-based therapeutics, such as cisplatin, has shown to have antitumor activity in cancers with DNA repair defects [33]. PARP1-*i* can also be used as a radiosensitizer [34]. Additionally, the combination of cDDP and PARP1-*i* proved to be potentially more effective in inhibiting cell proliferation more than both drugs apart [33]. BRCA1 or BRCA2 negative cancers cannot use the HR pathway to repair DNA DSBs. As a consequence, only cNHEJ is activated, which leads to chromosomal instability [30]. When using HT, and thus temporarily inhibiting the HR pathway, a BRCAness situation is mimicked in non-BRCA1 or BRCA2 mutated tumors. Therefore, patients who have tumors with active BRCA proteins can also benefit from PARP1-*i* treatments when combined with HT.

The combination of radiotherapy with chemotherapy or radiotherapy with hyperthermia are currently clinically applied to treat women with LACC. Considering the poor survival and high rates of severe side-effects in women with LACC, there is an urgent need to increase the effectiveness to improve tumor control without further challenging toxicity. The aim of our study was to develop a treatment modality that may overcome the limitations of current clinical treatments. In order to investigate this, different assays were executed to look at the effects on cell survival, DNA damage and cell death.

## 2. Results

### 2.1. Cellular Survival

The clonogenic assay was performed to study cell survival after various treatment combinations. cDDP alone significantly reduced the colony formation in all four cell lines. This effect was enhanced in all cell lines when PARP1-*i* was added as well. When combining 2 Gy of ionizing radiation (IR) with cDDP, this only showed a significant reduction in HeLa and CaSki cells compared to 2 Gy only. In all cell lines, adding PARP1-*i* led to a significant reduction compared to 2 Gy only. When comparing triple modality PARP1-*i*, 2 Gy and cDDP to standard treatment cDDP, together with 2 Gy, there was a significant reduction in SiHa and HeLa cells. The same effects were observed when cells were treated with 4 Gy. Adding PARP1-*i* to the standard chemoradiation treatment significantly reduced colony formation in cell lines SiHa and C33A (Figure 1A, Appendix A). HT alone did not significantly reduce colony formation compared to untreated cells in all four cell lines. When PARP1-*i* was added as well, the colony formation was significantly lower in all four cell lines. Combining 2 Gy with HT significantly lowered the colony formation in SiHa and C33A cells. The triple combination, IR, HT and PARP1-*i* showed a significant difference in colony formation compared to 2 Gy IR alone in all four cell lines. There was a significant reduction in colony formation when comparing 2 Gy IR, HT and PARP1-*i* with thermoradiation in SiHa and Caski cells. Cells treated with 4 Gy IR showed low normalized colony formation in all four cell lines, which was even significantly lower when combined with HT in SiHa, HeLa and C33A cells and even more so in the triple combination in these cell lines (Figure 1B).

As shown by the clonogenic survival curves in Figure 1, all cell lines demonstrate sensitivity to ionizing radiation, but none of the cell lines demonstrated synergistic cell death after adding cDDP to IR (Figure 1C, Appendix A). Synergism is the situation in which the combinational activity of two drugs exceed their expected additive or independent response [35]. However, adding PARP1-*i* to this combination lowered the surviving curve significantly, which was even further lowered in the quadruple treatment modality in all four cell lines (Figure 1C). HT only seems to enhance radiosensitivity, but the effect is not that pronounced. The triple combination IR + HT + PARP1-*i* is the most effective triple combination in all cell lines. In some cell lines (SiHa, CaSki, and C33A), this combination might be just as effective as the quadruple treatment. Only in the HeLa cells do the quadruple treatment seem to have more effect than this triple treatment at doses above 2 Gy (Figure 1C,D).

### 2.2. DNA Damage

The γ-H2AX assay was used to study the induction of DNA DSBs at 30 min after treatment (Figure 2A–C) and the amount of unrepaired DNA DSBs at 24 h after treatment. A single dose of IR (2 Gy) in SiHa, HeLa and CaSki cells already leads to significantly more foci compared to the other single therapies (Figure 2A,B, Appendix A). The single therapy PARP1-*i* showed no more foci compared to the control (Appendix A). IR + cDDP, the standard treatment in clinical practice, induced significantly fewer γ-H2AX foci after 30 min in SiHa cells compared to IR only. In HeLa cells, the induction of γ-H2AX foci was fewer after any treatment. In C33A cells the dual treatment had more effect than IR alone. However, though the number of foci increased when adding PARP1-*i* to the treatment, compared to IR alone in SiHa and C33A cells, the number of foci actually decreased in the HeLa and CaSki cells. In all cell lines, except C33A, the largest effect was found in the quadruple treatment (Figure 2A). The standard treatment IR + HT had more foci compared to IR only 30 min after treatment for SiHa, HeLa and C33A cells, but fewer for CaSki cells. In all cell lines, except for HeLa cells, the effect was enhanced when cells were treated with PARP1-*i* as well. Adding cDDP to this combination only had an additional enhancing effect in CaSki cells (Figure 2B). Representative pictures of a cell nuclei for all treatments were included in Figure 2C.

The effect of cDDP was more pronounced after 24 h (Figure 2D–F) than after 30 min. The number of foci in samples with cDDP was so high that single foci could not be distinguished from each other. The number of cells with this large number of foci was counted (Appendix A, right), but was found to be the same for all cDDP-containing samples. In SiHa cells ± 65% of cells individual foci were uncountable in cDDP-containing samples. In HeLa cells, this percentage was between 70% and 80% and in CaSki cells 60–65%. In C33A cells, there were 30–40% cells with uncountable individual foci (Figure 2D). In the samples treated with IR, HT and PARP1-*i*, the γ-H2AX foci could be separately distinguished and counted. There were fewer foci after 24 h compared to the samples after 30 min (Figure 2B,E). In SiHa, HeLa, and CaSki cells, there were more foci in the triple treatment IR + HT + PARP1-*i* compared to the single treatments and IR + HT after 24 h (Figure 2E, Appendix A). In CaSki cells, there were also more foci in IR treated and IR + HT treated cells compared to the other conditions. In C33A cells, there were almost no foci left 24 h after treatment in all samples (Figure 2E). Representative pictures are shown in Figure 2F. 

### 2.3. Cell Death

The live-cell imaging assay was used to visualize dying and dead cells 96 h and 136 h after treatment (Figure 3). Almost all dual and triple therapies had significantly more dying cells (Annexin V positive) in all cell lines compared to the control 96 h and 136 h after treatment (Appendix A). Comparing IR + cDDP with IR only shows no significant increase in dying cells 96 h after treatment. Only in CaSki cells was there a significant increase 136 h after treatment (Appendix A). The percentage dying cells is significantly higher at both time points in the triple combination IR + cDDP + PARP1-*i* compared with IR only in SiHa and CaSki cells. In all cell lines, the triple combination IR + cDDP + PARP1-*i* had more dying cells compared to standard treatment chemoradiation. This effect was significant 136 h after treatment in SiHa and CaSki cells (Figure 3A). The four modality treatment IR + HT + cDDP + PARP1-*i* showed the highest amount of dying cells in all samples. The HT-based graphs show the same trend. Triple modality IR + HT + PARP1-*i* has a higher percentage of dying cells compared to IR only and standard treatment thermoradiation. However, a significant increase was almost solely seen when comparing IR only or thermoradiation with the four modality treatment IR + HT + cDDP + PARP1-*i* (Figure 3B, Appendix A).

There were significantly more dead cells due to apoptosis and necrosis (YoYo3 positive) in almost all dual and triple therapies compared to the control in all cell lines both at 96 h and 136 h after treatment (Appendix A). Comparing standard treatment IR + cDDP with IR was only significantly different in CaSki cells. However, in the triple combination with PARP1-*i* there was a larger increase in cell death in all cell lines, although not always significant. Most significant differences could be found when comparing the four modality treatments with either IR only or IR + cDDP (Figure 3C). Additionally, in the HT-based graphs, most significant differences could be found when comparing the four modality treatment with either IR only or IR + HT (Figure 3D). Even though not all differences were significant, a clear trend could be observed in all cell lines, both in percentage of dying cells, as well as dead cells due to apoptosis and necrosis (Figure 3C,D). 

Additionally, when comparing the confluence, the same trend was present (Figure 3E,F). In all cell lines, the standard dual treatments had a confluence that was less compared to IR only. This effect was larger when combined with PARP1-*i* (Figure 3E,F and Appendix A). In all cell lines, the lowest confluence could be observed in the four-modality treatment (Figure 3E,F). Representative pictures are shown in Figure 3G.

## 3. Discussion

The radiobiological response to IR, cDDP, HT, PARP1-*i* and combinations in human cervical cancer cell lines vary per treatment and cell line. IR and cDDP both lead to cancer cell death, primarily by inducing DNA damage, whereas PARP1-*i*, HT and also cDDP inhibit or disturb the DNA repair pathways. Additionally, the trend of the effects is generally similar amongst different cervical cell lines; the clinical impact of the treatments is quantitatively different. These differences can be largely explained by the different working mechanisms of different treatments and the different cellular sensitivity to each of the drugs or modalities. 

The PARP1-inhibitor olaparib is used in the clinic to treat different types of cancer. Unfortunately, it is also associated with side effects such as nausea, rash and myalgias [36]. Up to 10% of patients suffer from severe side effects, such as grade 3 anemia, neutropenia and thrombocytopenia [30]. At the same time, suboptimal clinical effectiveness has been reported, which underlines the need for more effective PARP1-*i* treatment combinations, for instance with modalities like chemoradiation and thermoradiation. The benefit of such a combination can be a combination of enhanced effectiveness with acceptable side effects [37]. The advantage of combining systemic therapies like chemotherapy and PARP1-*i* with local therapies like ionizing radiation and hyperthermia is that the latter two will not enhance the systemic side effects of the former two, while at the same time the local tumor effects are enhanced as a result, which has been shown in the present study. 

Olaparib is mainly effective in cells in which BRCA1/2 is not functional, such as tumor that have BRCA1/2 mutations. Hyperthermia can temporarily downregulate levels of BRCA2, and thereby mimic a BRCAness situation. As a consequence, the combination of radiotherapy, which is local, local hyperthermia and Olaparib, are expected to cause minimal side effects, as only tumor cells have ineffective BRCA2, which are the main target of Olaparib. In contrast to this triple modality, chemotherapy, e.g., cisplatin, is a systemic treatment, and the combination with Olaparib can potentially cause severe side effects. However, this still needs to be explored thoroughly.

According to a paper by Bianchi et al. (2019), 0.15 µM and 1.5 µM of Olaparib are sufficient to inhibit PAR expression in cervical cancer cells after exposure for 24–48 h [38]. In the paper by Mann et al. (2019), the IC50 for the PARP1-*i* PJ34 was determined at approximately 10 µM for cervical cancer cell lines. When cells were exposed for approximately 48 h to the drug, this was already sufficient to inhibit PAR expression [39]. In this research, 4–5 µM of Olaparib, which is the determined IC50 for all cell lines, was used for a longer period of time, which is why sufficient inhibition of these proteins is suspected. A future step in our research is therefore to obtain additional molecular data to validate our study results regarding PARP1 expression.

Radiotherapy combined with cDDP is the standard treatment of care for patients with LACC. When patients have contraindications for cDDP, radiotherapy in combination with HT can be applied. We have shown, in this research, that the anticancer effects of both standard treatment combinations increase significantly when they are combined with PARP1-inhibition. By adding PARP1-*i* and applying a (modest) RT/cDDP dose reduction in those treatment regimens, we might achieve an improvement in the tumor control combined with a reduction of toxicity and occurrence of severe side effects. 

Besides PARP1-*i*, there are other targeted drugs under investigation in order to sensitize cervical cancer cells to cisplatin. Several agents targeting the epithelial growth-factor receptor (EGFR) are under investigation, since EGFR is overexpressed in 50–70% of cervical cancers. Cetuximab is the EGFR inhibitor of interest in several clinical trials. However, even though the drug cetuximab seems promising, in combination with cDDP and RT, this often leads to excessive toxicity and premature termination of the treatment [40]. Besides this, it might also be good to look into other targets. The PIK3CA gene, involved in the PI3K signaling pathway, is mutated in 13–36% of the cervical cancer cases, which is why it would be a good potential drug target. However, even though the in vitro data seems promising, more investigation is needed before it can be taken to the clinic [41]. Another target of interest may be JAK2, an activator of the transcription factor STAT3. When inhibiting JAK2 by using the clinically available inhibitor ruxolitinib, this leads to the inhibition of proliferation and induction of apoptosis in HPV positive cervical cancer cells. However, again, this is only supported by in vitro data, even though the drug is clinically available [42]. Therefore, this research focusses on the targeting drug PARP1-*i*, which shows promising results on the cellular level and is clinically available against cervical cancer.

Our research shows that addition of PARP1-*i* to either IR + cDDP or IR + HT has a significant effect on clonogenicity. When comparing, e.g., 4 Gy + cDDP to 2 Gy + cDDP + PARP1-*i* or 4 Gy + HT to 2 Gy + HT + PARP1-*i,* differences in cell survival (in all cell lines) are not significantly different. This might suggest that, by adding PARP1-*I,* the dose of ionizing radiation could be reduced without losing the effectiveness of treatment, or alternatively the effectiveness of ionizing radiation can be further enhanced without significantly enhancing the risk of serious side effects. In Figure 1C,D it is shown that the surviving fraction of the four modality treatment is slightly lower compared to PARP1-inhibition with chemoradiation, while there are no differences seen compared to the addition of PARP1-inhibtition to thermoradiation. In terms of DNA damage, there were no significant differences found between the four-modality treatment and any triple modality combination (Appendix A). In terms of DNA damage, in most cases there were no significant differences found between the triple modalities and the four-modality treatment. In apoptotic levels, however, in the majority of cell lines, the four-modality treatment does not induce significantly higher levels of apoptosis compared to either triple-modality treatment. Both radiosensitizers, cDDP and HT are effective when combined with IR, but this effect on the cervical cancer cells is enhanced when PARP1-*i* was added as well. This can also be seen in the γ-H2AX foci assay. The combinational treatments with PARP1-*i* were effective 30 min after treatment and also 24 h after treatment. When PARP1-*i* was given together with IR + HT or IR + cDDP, this resulted in more induction of DNA DSBs and less repair of the DSBs.

This is in concordance with the results of the live-cell imaging. When comparing the standard dual modalities with the addition of PARP1-*i*, there is an increase in percentage early apoptotic and late apoptotic and necrotic cells in all cell lines, indicating that these triple modalities eventually lead to more cell death and thus improve the standard treatments. We expected to see a larger difference in late apoptotic and necrotic cells, since IR induces not only apoptosis and necrosis, but also mitotic catastrophy and autophagy, which would all be visible by using the YoYo3 dye [43]. Research has also shown that PARP1-*i* is involved in the necrosis pathway [44]. However, based on our results, we do not see a decrease in necrosis when combining this with chemoradiation or thermoradiation. Therefore, this, combined with the increased induction of DSBs, indicates that PARP1 is more important in the DNA repair pathways than on the necrosis pathway. 

Cisplatin is more efficient with regard to DNA DSB formation in SiHa, HeLa and CaSki cell lines than in C33A cells. This might be due to the fact that the C33A cells are HPV negative and have mutated p53 expression [45]. Apart from that, it expresses only truncated pRb, which can be the reason why cNHEJ is less active. cDDP also acts upon MLH1, one of the key players in the SSB repair pathway Mismatch Repair (MMR). However, research has also shown that C33A cells express the smallest DNA content compared to other cells, which might also be a logical explanation why there are less γ-H2AX foci present in this cell line. Others reported similar observations [46]. In a previous publication, our group found a higher amount of foci per nucleus, particularly when cells were irradiated directly after HT treatment. In that study, it was chosen to exclude the cells in the S-phase of the cell cycle. Cells that were excluded had a low number of foci, because irradiation is minimally effective in this cell cycle phase [47]. In the analysis of the present study, it was logical to look at cells of all cell cycle phases, since cDDP mainly affects cells in the S-phase. There might be a cell cycle arrest in the chemoradiation group, which can result in the accumulation of DSBs, which cannot be repaired by the repair pathways. Previous research from our group had shown that Rad51, involved in HR, was no longer present after HT treatment, indicating that HR is no longer active [48]. With regard to the live-cell imaging results, there are some differences when comparing the different cell lines. Whereas SiHa and CaSki cells are both HPV-16 positive, they respond differently to the therapeutic agents used. CaSki cells are more sensitive to cDDP, whereas SiHa cells are more resistant to all treatment options. These findings were also supported by other research, especially regarding apoptotic pathways [49]. 

The samples containing cDDP contained more γ-H2AX foci after 24 h compared to 30 min, whereas the IR-only samples contained more γ-H2AX foci per nucleus after 30 min; this indicates that IR acts much faster in inducing DNA DSBs. These effects are enhanced when HT and/or PARP1-*i* are given as well. After 24 h, the effects of cDDP are more pronounced, indicating that cDDP acts relatively slowly as a DNA damaging agent. The effects of cDDP are pronounced, since many cells have an uncountable amount of foci. Other research groups also observed the many foci per nucleus, which is why they quantify cells in a similar way [50,51]. Moreover, we found clear foci when cervical cancers cells were exposed for a short period of time to cDDP [48,52]. In the current study, we had to expose cells for a long period to cDDP, which led to an uncountable amount of foci. These cells demonstrate a nuclear-wide γ-H2AX staining, because the cells experience replication stress caused by platinum-DNA adducts that prolong the S-phase of the cell cycle [53,54]. Besides, it is conceivable that a lot of cells already died after the combined effects of IR, HT and PARP1-*i*, and the population of surviving cells, managed to repair most of the DNA DSBs.

Combinational treatment of RT with HT and PARP1-*i* will possibly allow for a reduction of cDDP and systemic toxicity, without compromising or even improving tumor control. It has been demonstrated that the addition of PARP1-inhibitor to cDDP and HT allowed an almost ten-fold reduction of cDDP without diminishing the therapeutic effectiveness [48]. Moreover, adding PARP1-inhibitor to cDDP was found to be a promising approach to overcome cisplatin resistance and to achieve a better therapeutic effect [39]. However, it should be taken into account that treatments with PARP1-inhibitor were found to cause serious side effects in a clinical setting [55,56]. Therefore, finding the sweet spot of tolerable doses of cDDP and PARP1-inhibitor is the main challenge, which provides thorough research in in vivo studies and in clinical trials.

Adding PARP1-*i* to IR + cDDP or IR + HT might even lead to synthetic lethality, a phenomenon in which the combination of therapies leads to a strong synergistic effect rather than merely an additive effect [57,58]. When cells are treated with HT, this reduces the efficacy of HR, making the cells more susceptible to PARP1-inhibition [57,59]. Previous research has also shown that the standard treatment cDDP + HT can benefit from inhibition of PARP1 [48]. Our results are in concordance with these findings from literature, indicating that both standard treatments benefit from the addition of PARP1-*i*. Therefore, these results are a promising step towards novel treatment combinations for cervical cancer therapy.

## 4. Materials and Methods

### 4.1. Cell Cultures

Human cervical cancer cell lines with different HPV status, HeLa, SiHa, CaSki and C33A were obtained from the American Type Culture Collection (ATCC, Manassas, VA, USA). HeLa, SiHa and C33A cells were cultured in Eagle’s minimum essential medium containing 10% fetal bovine serum (FBS) and 2 mM glutamine. CaSki cells were cultured in Roswell Park Memorial Institute (RPMI) 1640 Medium supplemented with 10% heat-inactivated FBS and 2 mM glutamine. All cells were cultured at 37 °C in an incubator with humidified air supplemented with 5% CO_2_. The medium was supplied from Gibco-BRL life technologies, Breda, The Netherlands.

### 4.2. Irradiation

Cells were treated with ionizing radiation (IR) with γ-rays from a ^137^Cs source at a dose rate of about 0.5 Gy/min with multiple doses (0, 2, 4 and 6 Gy).

### 4.3. Hyperthermia

HT treatment was given at 42 °C for 1 h immediately prior to IR. HT was carried out by partially submerging the wells plates or dishes in a thermostatically-controlled water bath. The temperature was checked and the desired temperature (±0.1 °C) was reached in approximately 5 min.

### 4.4. Chemical Agents

Cisplatin was used as a chemotherapeutic agent, since this is often used in the clinic to treat cervical cancer. Cells were treated continuously with 0.3–0.5 µM cisplatin (cDDP, Platosin^®^, Pharmachemie B.V., Haarlem, The Netherlands) based on the measured IC50 depending on the cell line. 

PARP1-inhibition (PARP1-*i*) was induced using 4–5 µM olaparib (Lynparza^®^, AstraZeneca, Cambridge, UK), dissolved in medium continuously.

### 4.5. Clonogenic Assay

Clonogenic assays were performed in HeLa, SiHa, C33A and CaSki cells as described by Franken et al. (2006) [60]. Cells were plated prior to treatment at different densities in 6-wells plates. Both chemical agents, cDDP and PARP1-*i*, were given immediately when seeding cells, 4 h prior to the HT treatment and/or IR. After receiving all treatments, the cells were incubated for 10–12 days to form colonies. Surviving colonies were stained with a glutaraldehyde-crystal violet solution and counted manually. 

### 4.6. Immunocytochemistry

To investigate the amount of DSB breaks in HeLa, SiHa, C33A and CaSki cells after combinational treatments, the γ-H2AX foci assay was conducted. Cells were seeded at a density of 3.0 × 10^5^ cells on sterile coverslips in a 60 mm culture dish. Medium was added after 2 h to end up with a total volume of 3 mL per dish. After 24 h, cells were treated with cDDP and/or PARP1-*i*. After incubating for 30 min, the cells were treated with HT and/or IR. Approximately 30 min and 24 h after treatment, cells were fixated with 2% paraformaldehyde (PFA). After fixation, cells were permeabilized with PBS supplemented with 0.1% Triton X-100 and 1% FGS (TNBS) at room temperature. Thereafter, the cells were stained with primary mouse monoclonal antibody anti-γ-H2AX (Merck Millipore, Darmstadt, Germany, 1:100 in TNBS) and secondary antibody goat-anti-mouse-Cy3 (Jackson-Immunoresearch, Ely, Cambridgeshire, UK, 1:100 in TNBS) for 90 min and 30 min, respectively. DAPI staining (Life technologies, Carlsbad, CA, USA) was used to stain the nuclei. Vectashield with DAPI was used and foci scoring was done using a fluorescence microscope (Leica, Wetzlar, Germany). The amount of γ-H2AX foci formed per nucleus was examined manually.

### 4.7. Live-Cell Imaging

Live-cell imaging was performed using the IncuCyte S3 Live-cell analysis system (Essen BioScience, Ann Arbor, MI, USA). Cells were seeded in a 24-wells plate at a density of 2.500–4.000 cells. Cells were treated 24 h later with cDDP, PARP1-*i*, HT and IR. After treatment, the medium was refreshed for medium supplemented with Annexin V (AdipoGen Life Sciences, San Diego, CA, USA) and YoYo3 (Fisher Scientific, Waltham, MA, USA) to visualize early apoptotic (Annexin V-positive) and late apoptotic/necrotic (YoYo3-positive) cells. The 24-wells plate was placed on the microscope stage in the incubator chamber in 5% CO_2_ at 37 °C. The Images were automatically acquired at different time intervals for 6 days. The data were processed using IncuCyte image analysis software (IncuCyte S3 Software, Essen BioScience Ann Arbor, MI, USA). 

### 4.8. Statistical Analyses

Statistical tests on data were based on an independent samples *t*-test and the Mann Whitney *U* test.

## 5. Conclusions

Combination treatment with PARP1-*i* enhances both standard treatment options IR + cDDP and IR + HT in cervical cancer cell lines. Surviving fractions are lower, and effects on DNA DSBs last longer. The amount of dying and dead cells is increased when PARP1-*i* is given in combination with both standard treatments. The effect of PARP1-*i* on surviving fractions suggests that PARP1-*i* sensitizes cells for the conventional therapies. This conclusion was confirmed by the γ-H2AX data 24 h after treatment. However, the results of the live cell imaging suggest that effects are solely additive, meaning that the result of two drugs act together as the sum of the effects when the drugs are given independently [35]. When thermoradiation or chemoradiation will be combined with PARP1-*i*, the on target cytotoxic effects will be enhanced, while the toxicity levels will remain manageable, indicating that combining the standard treatments for cervical cancer with PARP1-*i* will be a potentially effective treatment of locally advanced cervical cancer. 

## Figures and Tables

**Figure 1 cancers-13-02092-f001:**
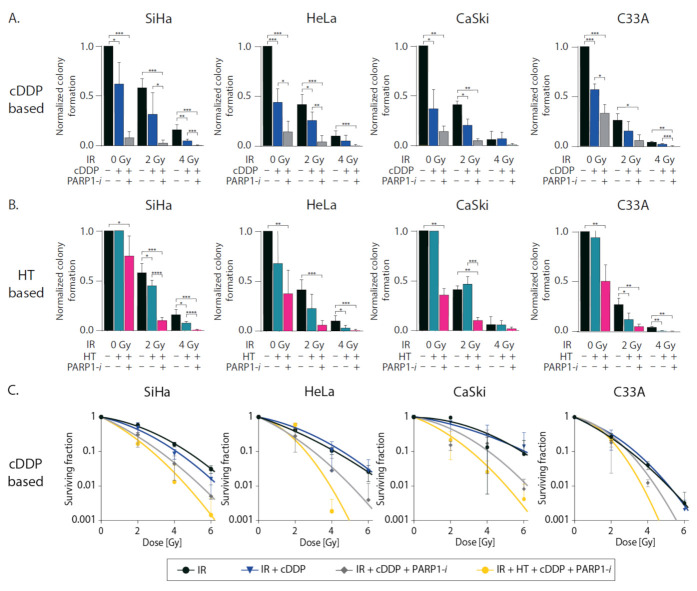
In vitro response to ionizing radiation (IR), cisplatin (cDDP), hyperthermia (HT) and PARP1-inhibition (PARP1-*i*) as single and combinational treatments in four cervical cancer cell lines (SiHa, HeLa, CaSki and C33A). (**A**) Normalized colony formation after IR in combination with cDDP or (**B**) with HT. (**C**) Clonogenic survival graph after IR in combination with cDDP or (**D**) with HT. * *p* < 0.05, ** *p* < 0.01, *** *p* < 0.001, **** *p* <0.0001 compared to corresponding controls. Independent samples *t*-test.

**Figure 2 cancers-13-02092-f002:**
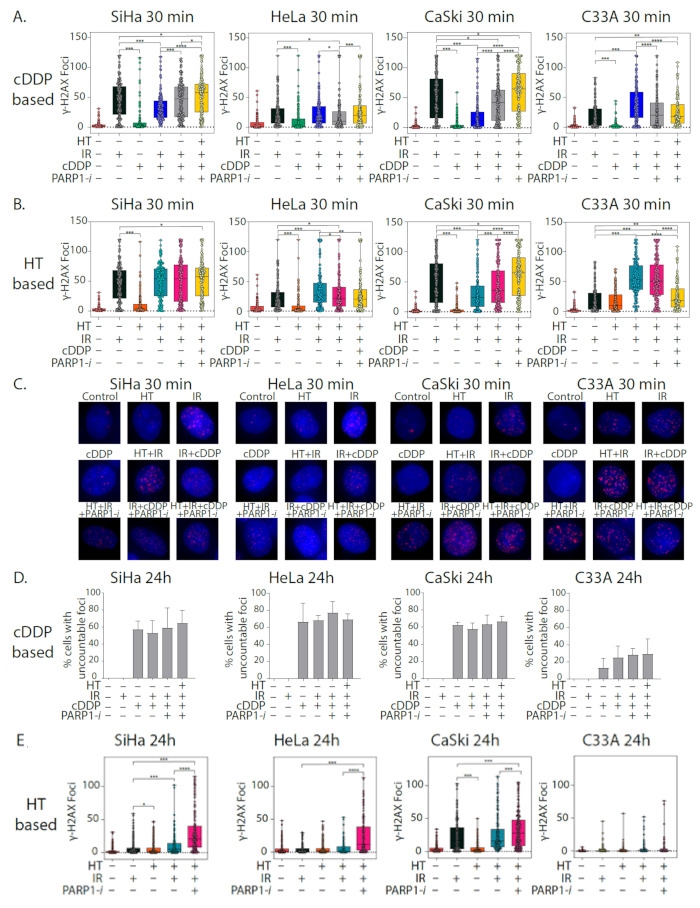
In vitro analysis of γ-H2AX foci per nucleus in cervical cancer cell lines SiHa, HeLa, CaSki, and C33A. (**A**,**B**) Displaying the effects 30 min after treatment with ionizing radiation (IR), cisplatin (cDDP), hyperthermia (HT) and PARP1-inhibition (PARP1-*i*) in different combinations. (**C**) Appearance of γ-H2AX foci in representative cervical cancer cell nuclei per treatment at 30 min after treatment. (**D**,**E**) Displaying the effects 24 h after treatment with ionizing radiation (IR), cisplatin (cDDP), hyperthermia (HT) and PARP1-inhibition (PARP1-*i*) in different combinations. (**F**) Appearance of γ-H2AX foci in representative cervical cancer cell nuclei per treatment at 24 h after treatment. * *p* < 0.05, ** *p* < 0.01, *** *p* < 0.001, **** *p* < 0.0001 compared with corresponding controls, Mann Whitney *U* test.

**Figure 3 cancers-13-02092-f003:**
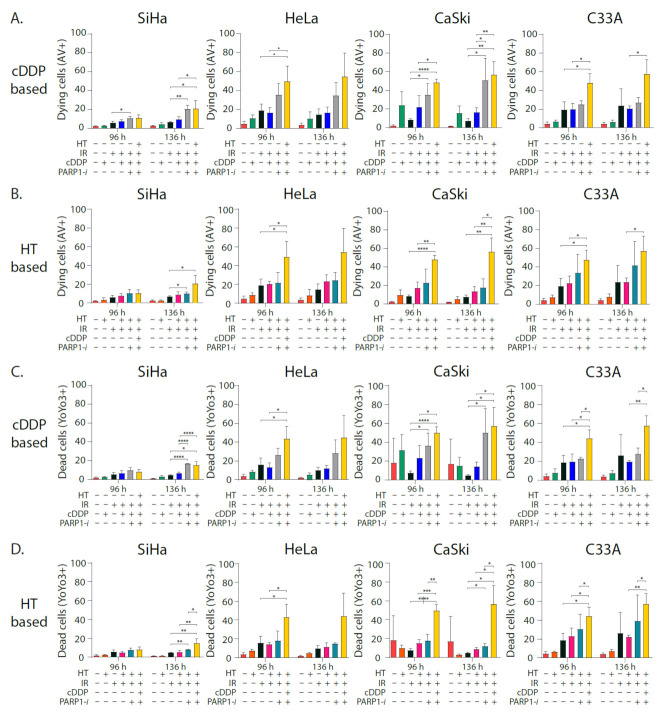
In vitro analysis of dying and dead cells in cervical cancer cell lines SiHa, HeLa, CaSki, and C33A. (**A**,**B**) Displaying the percentage of dying cells after treatment with 2 Gy ionizing radiation (IR), cisplatin (cDDP), hyperthermia (HT) and PARP1-inhibition (PARP1-*i*) in different combinations. (**C**,**D**) Displaying the percentage of dead cells after treatment with ionizing radiation (IR), cisplatin (cDDP), hyperthermia (HT) and PARP1-inhibition (PARP1-i) in different combinations. (**E**,**F**) Displaying the confluence per treatment normalized to the control. (**G**) Appearance of dying cells (AV+) and dead cells (YoYo3+) in HeLa cells per treatment at 96 h and 136 h after treatment. * *p* < 0.05, ** *p* < 0.01, *** *p* < 0.001, **** *p* < 0.0001 compared with corresponding controls, Independent samples *t*-test.

## Data Availability

The data presented was created and analyzed in this study.

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
