# Peer review of "PARP1-Inhibition Sensitizes Cervical Cancer Cell Lines for Chemoradiation and Thermoradiation"

_cancers, 2021, doi:10.3390/cancers13092092_

Round 1

Reviewer 1 Report

The paper is of interest to the scientific community and has implications for alterations and improvement to clinical practice, achieving greater tumor cell killing with combinations identified in these studies. The findings could indicate alternative therapeutic approaches - though there may be some contesting of the suggestion of dose reduction giving the same impact as there is long standing resistance amongst clinical practice to reduce maximal dosing that offers greatest cell killing. It may however be of interest to adapt the claim to be more effective on target with only similar levels of collateral toxicity (manageable levels), and perhaps removal of cDDP from a combination if proven to not be an advantage. 
Moving to more details of the paper, there are some typographical issues and layout which should be addressed but focus is with the scientific detail. Starting on line 92, PARP1i is acceptable as this is the predominant isoform target but it may be better and easier to explain with just PARPi – primarily because poly (ADP-ribose) polymerase is the family, not PARP1 specifically. This paragraph may thus benefit from beginning with full title and then PARPi or perhaps PARP1i thereafter. Line 97 platinumbased. Line 107 combination of radiotherapy. Line 130 ‘alone’ not ‘only’
Line 141 triple combination may be most effective but I think more could be made of the ability to focus these therapies to the tumor area (HT) and tumor selectively with IR thus further eliminating the off-target toxicity – PARPi may still lead to limitations of dose with haem tox but this would not be in conjunction with chemo tox from cDDP (which is in turn exacerbated by PARPi thus the 4 combined is much less compelling). Here is where thinking regarding dose reduction came to my mind though it is mentioned later – there may not be potential to reduce any therapeutic dose (particularly in well-established radiotherapy regimen) but limiting tox to the targeted area is a clear advantage.
Line 162 – fewer foci with cDDP vs IR alone – there should be more exploration of this effect in hypotheses regarding cycling cells – there are no cycle data in the paper unlike in the groups other publication (Mei et al 2020) – this is the first I will mention this paper but it is curious that it is not cited. There is a potential impact in cell cycle arrest from cDDP that in combination with IR may result in the cells being effectively stopped in their tracks and thus not processing DNA damage (unlikely to prevent damage itself but as foci are formed through attempts to rectify damage there is a clear potential overlap in the mechanism that I believe is brought out in the earlier work).
Line 202 life-cell to live-cell
H2AX foci – the consequence of cDDP exposure is ‘so high, that single foci could not be distinguished from each other.’ The role of H2AX is explored in this paper as DNA damage and is usually considered this way. However, there are many publications exploring H2AX in replication stress, a key consequence of cDDP exposure and it is entirely plausible that this is the cause of the observation here. An example that could be read and included in the description of both results and discussion is this recent Cancers paper from Moeglin et al : Uniform Widespread Nuclear Phosphorylation of Histone H2AX Is an Indicator of Lethal DNA Replication Stress – this appears to be a similar phenotype to the observation in Fig 2D. The data for the triple combination (HT, IR, PARPi) are largely well explained but there is an issue with the C33A data, particularly in consideration of the data in Mei et al 2020 – this seems inconsistent with those data in the same cell line – H2AX foci were reported to be observed at 24h and yet are completely resolved here.
Figure 3 – needs to be revised and results described for 3G – there is no mention of the yoyo data in this section.
Line 291 – C33A cannot ‘often’ have mutated p53 – they do or they do not. Tumors that are not HPV may often possess other mutations in p53 response and C33A are an example of such might be what is meant? Only hypophosphorylated Rb suggests a failure in kinases in these cells but is it just truncated Rb? Line 292 implies no cNHEJ but this is not clearly explained (although could explain lack of repair still not clear if explains lack of H2AX at 24h (in fact may be counterintuitive if no H2AX is because all cells are repaired)). Line 296 – others report similar observations – what about own work? There is some inconsistency here.
Materials and Methods
Given variation in the cell line data between publications, is it certain that these cells are as stated – STR profiles should be stated if completed. Hyperthermia – how was 42C selected – explained by citing Mei et al 

Conclusions are very brief – clear but there are other alternative explanations as above and there is some weakness in a proposal to give lower side effects without covering the clinical challenges – flipping the argument that more effective tumor kill for the same dose / toxicity improves the therapeutic index is one option.

Author Response

Point-to-point reply to the reviewers comments of IJff et al. “PARP1-inhibition enhances chemoradiation and thermoradiation in cervical cancer cells”, manuscript no. 1130089

We would like to thank the expert reviewers for critically evaluating our manuscript and providing us with valuable criticism that has helped us to improve the quality of our work.

We have been able to address all points that have been raised in this review process. 

Please find below an overview of our response to the issues raised as well as how we have addressed them in our revised manuscript:

Reviewer 1

The paper is of interest to the scientific community and has implications for alterations and improvement to clinical practice, achieving greater tumor cell killing with combinations identified in these studies. The findings could indicate alternative therapeutic approaches - though there may be some contesting of the suggestion of dose reduction giving the same impact as there is long standing resistance amongst clinical practice to reduce maximal dosing that offers greatest cell killing. It may however be of interest to adapt the claim to be more effective on target with only similar levels of collateral toxicity (manageable levels), and perhaps removal of cDDP from a combination if proven to not be an advantage. 

Moving to more details of the paper, there are some typographical issues and layout which should be addressed but focus is with the scientific detail.

We thank the reviewer for his/her kind words. With his/her permission, we wish to use these suggestions in our discussion and conclusion, since we agree that this would be a benefit for our paper.

  • Starting on line 92, PARP1i is acceptable as this is the predominant isoform target but it may be better and easier to explain with just PARPi – primarily because poly (ADP-ribose) polymerase is the family, not PARP1 specifically. This paragraph may thus benefit from beginning with full title and then PARPi or perhaps PARP1i thereafter.

We thank the reviewer for these insights. We have adjusted paragraph 4 from the introduction to: ‘PARP-inhibitors (PARP-i), e.g. olaparib are therapeutic agents targeting poly(ADP-ribose) polymerase, that also gained interest in the field of cervical cancer therapy. One of the key enzymes is Polymerase 1 (PARP1), which is nvolved in AltNHEJ. PARP1 is also involved in single-strand break (SSBs) repair. Olaparib is a clinically used PARP1-i chemotherapeutic agent which is especially effective for treatments of BRCA1 or BRCA2 negative cancers, such as ovarian and breast cancer types. PARP1-i alone or in combination with platinum-based therapeutics such as cisplatin has shown to have anti-tumor activity in cancers with DNA repair defects. PARP1-i can also be used as a radiosensitizer.’

  • Line 97 platinumbased.

Line 107 combination of radiotherapy.

Line 130 ‘alone’ not ‘only’

Our apologies for these typos, we have changed it to:

Line 97 ‘platinum based’

Line 107 ‘combination of’

Line 130 ‘alone’

  • Line 141 triple combination may be most effective but I think more could be made of the ability to focus these therapies to the tumor area (HT) and tumor selectively with IR thus further eliminating the off-target toxicity – PARPi may still lead to limitations of dose with haem tox but this would not be in conjunction with chemo tox from cDDP (which is in turn exacerbated by PARPi thus the 4 combined is much less compelling). Here is where thinking regarding dose reduction came to my mind though it is mentioned later – there may not be potential to reduce any therapeutic dose (particularly in well-established radiotherapy regimen) but limiting tox to the targeted area is a clear advantage.

We fully agree with the reviewer, and as mentioned above in answer to the introductory remark, we will add paragraph 10 to the discussion: ‘Combinational treatment of RT with HT and PARP1-i will possibly allow reduction of cDDP and systemic toxicity, without compromising or even improving tumor control. It has been demonstrated that the addition of PARP1-inhibitor to cDDP and HT allowed an almost ten-fold reduction of cDDP without diminishing the therapeutic effectiveness. Moreover, adding PARP1-inhibitor to cDDP was found to be a promising approach to overcome cisplatin resistance and to achieve a better therapeutic effect. However, it should be taken into account that treatments with PARP1-inhibitor were found to cause serious side-effects in clinical setting. Therefore, finding the sweet spot of tolerable doses of cDDP and PARP1-inhibitor is the main challenge, which provides thorough research in in vivo studies and in clinical trials.’

  • Line 162 – fewer foci with cDDP vs IR alone – there should be more exploration of this effect in hypotheses regarding cycling cells – there are no cycle data in the paper unlike in the groups other publication (Mei et al 2020) – this is the first I will mention this paper but it is curious that it is not cited. There is a potential impact in cell cycle arrest from cDDP that in combination with IR may result in the cells being effectively stopped in their tracks and thus not processing DNA damage (unlikely to prevent damage itself but as foci are formed through attempts to rectify damage there is a clear potential overlap in the mechanism that I believe is brought out in the earlier work).

We thank the reviewer for this suggestion. We added the following in paragraph 8 from the discussion: ‘In a previous publication our group found a higher amount of foci per nucleus, particularly when cells were irradiated directly after HT treatment. In that study it was chosen to exclude the cells in S-phase of the cell cycle. Cells that were excluded had a low number of foci, because irradiation is minimally effective in this cell cycle phase. In the analysis of the present study, we included cells of all cell cycle phases, since cDDP mainly affects cells in S-phase. There might be a cell cycle arrest in the chemoradiation group, which can result in accumulation of DSBs which cannot be repaired by the repair pathways.’

  • Line 202 life-cell to live-cell

Our apologies, this has been changed.

  • H2AX foci – the consequence of cDDP exposure is ‘so high, that single foci could not be distinguished from each other.’ The role of H2AX is explored in this paper as DNA damage and is usually considered this way. However, there are many publications exploring H2AX in replication stress, a key consequence of cDDP exposure and it is entirely plausible that this is the cause of the observation here. An example that could be read and included in the description of both results and discussion is this recent Cancers paper from Moeglin et al : Uniform Widespread Nuclear Phosphorylation of Histone H2AX Is an Indicator of Lethal DNA Replication Stress – this appears to be a similar phenotype to the observation in Fig 2D.

The reviewer is absolutely right. There are many articles that indicate whether cells are γ-H2AX positive or negative. We decided to count all the foci separately in order to get more insight on the number of breaks that remained unrepaired after the various treatments and inhibitors of DNA repair pathways. Thank you for referring to the article by Moeglin at all. This is indeed similar to our findings. Subsequently, we included a few lines in paragraph 9 from the discussion section: ‘Moreover, we found clear foci when cervical cancers cells were exposed for a short period of time to cDDP. In the current study, we had to expose cells for a long period to cDDP, which led to an uncountable amount of foci. These cells demonstrate a nuclear-wide γ-H2AX staining, because the cells experience replication stress caused by platinum-DNA adducts that prolong the S-phase of the cell cycle.’.

  • The data for the triple combination (HT, IR, PARPi) are largely well explained but there is an issue with the C33A data, particularly in consideration of the data in Mei et al 2020 – this seems inconsistent with those data in the same cell line – H2AX foci were reported to be observed at 24h and yet are completely resolved here.

 In the research published by Mei et al 2020, S-phase cells were excluded. In this phase of the cell cycle, most cells had a very low number of foci. This is also in line with the fact that IR had the most effect on cells which are in G2/M-phase. In the current research, cells from all phases were included, as a large focus on our study is focused on IR and cDDP DNA damage. For the latter, cells in S-phase are mostly targeted. As mentioned in point 1.4, we added a paragraph. In this paragraph, the paper by Mei et al. 2020 is cited as well.

  • Figure 3 – needs to be revised and results described for 3G – there is no mention of the yoyo data in this section

Our apologies that Figure 3 was not completely clear. To this end, we decided to change the legends of the figure. Instead of using ‘% early apoptotic cells’, we now use the description: ‘dying cells’. All dead cells, formerly known as ‘% late apoptotic/necrotic cells’, are considered to be due to apoptosis, necrosis and all other forms of cell death. Using these more general terms, we agree it is clearer for the reader.

In the text Figure 3G is added to the results section 2.3, paragraph 3.

  • Line 291 – C33A cannot ‘often’ have mutated p53 – they do or they do not. Tumors that are not HPV may often possess other mutations in p53 response and C33A are an example of such might be what is meant? Only hypophosphorylated Rb suggests a failure in kinases in these cells but is it just truncated Rb? Line 292 implies no cNHEJ but this is not clearly explained (although could explain lack of repair still not clear if explains lack of H2AX at 24h (in fact may be counterintuitive if no H2AX is because all cells are repaired)).

The reviewer is absolutely right, the C33A cells have mutated p53, which is what we meant to say, as well as truncated Rb instead of hypophosphorylated Rb. We adapted the text in paragraph 8 from the discussion: ‘This might be due to the fact that the C33A cells are HPV negative and have mutated p53 expression. Apart from that, it expresses only truncated pRb, which can be the reason why cNHEJ is less active.’

  • Line 296 – others report similar observations – what about own work? There is some inconsistency here.

The research published by Mei et al 2020, S-phase cells were excluded. In this phase of the cell cycle, most cells will be found with a very low number of foci. In our research, we included cells from all phases of the cell cycle due to the treatments that were investigated in this study.

  • Materials and Methods. Given variation in the cell line data between publications, is it certain that these cells are as stated – STR profiles should be stated if completed. Hyperthermia – how was 42C selected – explained by citing Mei et al.

All cell lines have recently been tested on HPV copy number and HPV status (unpublished results). They all have the correct HPV status and copy number according to literature. Therefore we are confident the cells lines are as stated. Moreover, mycoplasma tests are ran in our lab monthly, and all cells lines were negative during times of experiments for this manuscript.

  • Conclusions are very brief – clear but there are other alternative explanations as above and there is some weakness in a proposal to give lower side effects without covering the clinical challenges – flipping the argument that more effective tumor kill for the same dose / toxicity improves the therapeutic index is one option.

If you allow us, we will slightly adapt your wonderful text and add this to our conclusion: ‘When thermoradiation or chemoradiation will be combined with PARP1-i, the on target effects will be bigger, while the toxicity levels will remain manageable, indicating that combining the standard treatments for cervical cancer with PARP1-i will be a potentially effective treatment of locally advanced cervical cancer.’

Reviewer 2 Report

Ijff and colleagues studied the effect of PARP1 inhibitor (Olaparib) with or without Cisplatin / irradiation (IR) / hyperthermia (HT) in cervical cancer cells. Authors have used HPV-positive and HPV-negative cervical cancer cell lines (SiHa, CaSki, HeLa and C33A) and analyzed the cytotoxic effects of PARP1 inhibitor in combination with chemoradiation and HT. Chemoradiation-induced DNA damage leads to hyperactivation of PARP1 resulting in massive consumption of NAD+ for DNA repair. The current manuscript has merits however, the manuscript is not providing any insight whether PARP-1 inhibition induces irreversible cytotoxic effects upon chemoradiation and HT. I have few concerns mentioned below.
The results of the manuscript are not demonstrating that whether the PARP1 inhibition enhances the sensitivity chemoradiation and HT in HPV+ and HPV- cervical cancer cells or has an additive cytotoxic effect? The title of the manuscript should be modified as per their mechanism of action?  
Double standard DNA damage leads to PAR formation and undergoes cell necrosis. Did the author observe any PARylation product? 
The authors haven't provided the data showing the effect of combined chemoradiation and HT treatment with PARP-1 inhibitor on change in PARP-1 or on cleaved PARP expression? The authors need to provide the basal expression of PARP-1 and quantitative inhibition of PARP1 expression upon olaparib with or without chemoradiation and HT treatment? 
The effect of PARP-1 inhibition needs to be validated by silencing of PARP-1 expression followed with chemoradiation and HT treatment. 
Authors need to provide the irradiation D0 and SF2 (survival fraction 2) for all four cell lines. 
The conclusion should be placed before the method section. 

Line 388; authors need to remove "less side effects" since there is no evidanace has been provided to conclude this remark.  

Author Response

Point-to-point reply to the reviewers comments of IJff et al. “PARP1-inhibition enhances chemoradiation and thermoradiation in cervical cancer cells”, manuscript no. 1130089

We would like to thank the expert reviewers for critically evaluating our manuscript and providing us with valuable criticism that has helped us to improve the quality of our work.

We have been able to address all points that have been raised in this review process. 

Please find below an overview of our response to the issues raised as well as how we have addressed them in our revised manuscript:

 Reviewer 2

Ijff and colleagues studied the effect of PARP1 inhibitor (Olaparib) with or without Cisplatin / irradiation (IR) / hyperthermia (HT) in cervical cancer cells. Authors have used HPV-positive and HPV-negative cervical cancer cell lines (SiHa, CaSki, HeLa and C33A) and analyzed the cytotoxic effects of PARP1 inhibitor in combination with chemoradiation and HT. Chemoradiation-induced DNA damage leads to hyperactivation of PARP1 resulting in massive consumption of NAD+ for DNA repair. The current manuscript has merits however, the manuscript is not providing any insight whether PARP-1 inhibition induces irreversible cytotoxic effects upon chemoradiation and HT. I have few concerns mentioned below.

We thank the reviewer for thoroughly evaluating our manuscript and providing us with critical questions in order to improve our manuscript. The reviewer is correct that in this manuscript we did not discuss whether we can discern the additive and sensitizing effects of PARP1-i combined with chemoradiation and HT based on our results.

We added paragraph 3 in the discussion to elaborate on this point: ‘Olaparib is mainly effective in cells in which BRCA1/2 is not functional, such as tumor that have BRCA1/2 mutations. Hyperthermia can temporarily downregulate levels of BRCA2, and thereby mimic a BRCAness situation. As a consequence, the combination of radiotherapy, which is local, local hyperthermia and Olaparib, are expected to cause minimal side effects, as only tumor cells have ineffective BRCA2 which are the main target of Olaparib. In contrast to this triple modality, chemotherapy, e.g. cisplatin, is a systemic treatment, and the combination with Olaparib can potentially cause severe side effects. However, this still needs to be explored thoroughly.’

2.1.    The results of the manuscript are not demonstrating that whether the PARP1 inhibition enhances the sensitivity chemoradiation and HT in HPV+ and HPV- cervical cancer cells or has an additive cytotoxic effect?

That is a very relevant question. Therefore, we added the following text to our conclusion: ‘The effect of PARP1-i on surviving fractions suggests PARP1-i sensitizes cells for the conventional therapies. This conclusion was confirmed by the γ-H2AX  data 24h after treatment. However, the results of the live cell imaging suggests effects are solely additive.’

2.2.    The title of the manuscript should be modified as per their mechanism of action?  

Thank you for this suggestion. We agreed that the title would be stronger when adjusting it to: ‘PARP1-inhibition sensitizes cervical cancer cell lines for chemoradiation and thermoradiation’.

2.3.    Double standard DNA damage leads to PAR formation and undergoes cell necrosis. Did the author observe any PARylation product? 

In this study, we focused on cell survival, formation and repair of DNA double strand breaks, and live cell imaging to determine apoptosis and necrosis. We did not study the PARylation product. However, we found that Bianchi et al used a concentration of 0.15-1.5uM for and expose time of 24-48h in cervical cancer cells, and they found very nice inhibition of PAR expression. Moreover, Mann et al, used the IC50 of PJ34, which was found to be 10uM at an expose period of 48 h on cervical cancer cells. They also found complete PAR inhibition. As we used 4-5uM, which was determined as the IC50 in our cell lines, and we exposed our cells for a longer period of time to the PARP1-inhibitor. Based on the previous findings, we expect to have sufficient inhibition of these proteins.

2.4.    The authors haven't provided the data showing the effect of combined chemoradiation and HT treatment with PARP-1 inhibitor on change in PARP-1 or on cleaved PARP expression? The authors need to provide the basal expression of PARP-1 and quantitative inhibition of PARP1 expression upon olaparib with or without chemoradiation and HT treatment? 

The reviewer is correct that we did not emphasize the effect of the four-modality treatment. We choose to highlight the addition of PARP1-inhibition to the currently standard treatments in the clinical, which are chemoradiation and thermoradiation. However, we did perform the four-modality treatment in all experiments. As a response, we added some text in paragraph 6 from the discussion: ‘In Figure 1C and D it is shown that the surviving fraction of the four-modality treatment is slightly lower compared to PARP1-inhibition with chemoradiation, while there are no differences seen compared to the addition of PARP1-inhibtition to thermoradiation. In terms of DNA damage, there were no significant differences found between the four-modaility treatment and any triple modality combination (Supplementary Table 2). In apoptotic levels, however, in the majority of cell lines, the four-modality treatment does not induce significantly higher levels of apoptosis  compared to than either triple modality treatment.’

2.5.    The effect of PARP-1 inhibition needs to be validated by silencing of PARP-1 expression followed with chemoradiation and HT treatment. 

The reviewer raises an important point. According to a paper of Bianchi et al, Gynecol ONcol 2019, 0.15uM and 1.5uM of Olaparib are sufficient in inhibiting PAR expression in cervical cancer cells after exposure of 24 h – 48 h. In the paper of Mann et al Oncotarget 2019, the IC50 for the PARP1-i PJ34 was determined at approximally 10 uM for cervical cancer cell lines. When cells were exposed for approximately 48h to the drug, this was already sufficient to inhibit PAR expression. As we used 4-5uM of Olaparib for a longer period of time, which is the determined IC50 for all cell lines, we expect to have sufficient inhibition of these proteins.

 2.6.    Authors need to provide the irradiation D0 and SF2 (survival fraction 2) for all four cell lines. 

Our apologies for not including this information. An additional supplementary table (Supplementary table 4) is added to our manuscript containing the plating efficiency and the surviving fraction per cell line.

2.7.    The conclusion should be placed before the method section. 

The template provided by Cancers was used which prescribes a particular order of sections, but we agree with the reviewer that it is indeed more logical to place the conclusions before the method section to get a better flow while reading. We have changed the order accordingly.

2.8.    Line 388; authors need to remove "less side effects" since there is no evidence has been provided to conclude this remark.  

Our apologies. We adapted the text in the conclusion.

 Reviewer 3 Report

In this manuscript,  IJff et al. investigate the potential of PARP inhibition as a method to enhance the response of cervical cancer cells to chemoradiation and thermoradiation. Whilst the manuscript is interesting and demonstrates a potential for this treatment regime, the authors should make their data presentation and should place their results in the context with the currently available literature.

  • All Figure - all of the graphs have text that is too small and is almost illegible. Furthermore, the colours on the graph should be removed as it is very distracting and would be inappropriate for people with vision problems (the graphs are labelled well enough without the colours).
  • Figure 1A - the bottom of the graph has been cut off.
  • Figure 2 - do the yH2AX foci overlap with foci repair factors such as Rad51 or 53BP1? This would be particularly interesting to look at at 24h to show that the high level of yH2AX in the combo treatments corresponds to unrepairable DNA.
  • Figure 3G - this is not referenced in the results or the figure legend.
  • More discussion should be give on other methods to sensitize cervical cancer cells to cisplatin (such as Arjumand et al., PMID: 27489350 (PI3K inhibitor); Moore et al., PMID: 22960004 (EGFR inhibitor); Morgan et al., PMID: 31817106 (JAK2 inhibitor)).

Author Response

Point-to-point reply to the reviewers comments of IJff et al. “PARP1-inhibition enhances chemoradiation and thermoradiation in cervical cancer cells”, manuscript no. 1130089

We would like to thank the expert reviewers for critically evaluating our manuscript and providing us with valuable criticism that has helped us to improve the quality of our work.

We have been able to address all points that have been raised in this review process. 

Please find below an overview of our response to the issues raised as well as how we have addressed them in our revised manuscript:

 Reviewer 3

 In this manuscript, IJff et al. investigate the potential of PARP inhibition as a method to enhance the response of cervical cancer cells to chemoradiation and thermoradiation. Whilst the manuscript is interesting and demonstrates a potential for this treatment regime, the authors should make their data presentation and should place their results in the context with the currently available literature.

  • All Figure - all of the graphs have text that is too small and is almost illegible. Furthermore, the colours on the graph should be removed as it is very distracting and would be inappropriate for people with vision problems (the graphs are labelled well enough without the colours).

We apologize for this inconvenience. We muted the colours and made the text in each figure bigger.

  • Figure 1A - the bottom of the graph has been cut off.

Our apologies, apparently something went wrong when making a merged PDF. The editorial system did not permit us to not check this file when submitting. Thank you for spotting this error. This has been changed. To make sure this mistake will not be made again, we also uploaded a PDF version of our manuscript.

  • Figure 2 - do the γ-H2AX foci overlap with foci repair factors such as Rad51 or 53BP1? This would be particularly interesting to look at at 24h to show that the high level of yH2AX in the combo treatments corresponds to unrepairable DNA.

Earlier research published by our group has looked into Rad51 foci (Oei et al. Oncotarget, 2017). Rad51 was no longer present after HT treatment, indicating that HR is no longer active. We added this to paragraph 8 from the discussion  ‘Research from our group had previously shown that Rad51, involved in HR, was no longer present after HT treatment, indicating that HR is no longer active.’

  • Figure 3G - this is not referenced in the results or the figure legend.

Our apologies, we adjusted the text in the legend and the results section.

  • More discussion should be given on other methods to sensitize cervical cancer cells to cisplatin (such as Arjumand et al., PMID: 27489350 (PI3K inhibitor); Moore et al., PMID: 22960004 (EGFR inhibitor); Morgan et al., PMID: 31817106 (JAK2 inhibitor)).

 The reviewer is right, so we added paragraph 5 to the discussion: ‘Besides PARP1-i, there are other targeted drugs under investigation in order to sensitize cervical cancer cells to cisplatin. Several agents targeting the epithelial growth-factor receptor (EGFR) are under investigation, since EGFR is overexpressed in 50-70% of the cervical cancers. Cetuximab is the EGFR inhibitor of interest in several clinical trials. However, even though the drug cetuximab seems promising in combination with cDDP and RT, this often leads to excessive toxicity and premature termination of the treatment. Besides this, it might also be good to look into other targets. The PIK3CA gene, involved in the PI3K signaling pathway, is mutated in 13-36% of the cervical cancer cases, which is why it would be a good potential drug target. However, even though the in vitro data seems promising, more investigation is needed before it can be taken to the clinic. An-other target of interest may be JAK2, an activator of the transcription factor STAT3. When inhibiting JAK2 by using the clinically available inhibitor ruxolitinib, this leads to the in-hibition of proliferation and induction of apoptosis in HPV positive cervical cancer cells. However, again, this is only supported by in vitro data, even though the drug is clinically available. Therefore, this research focusses on the targeting drug PARP1-i, which shows promising results on cellular level and is clinically available against cervical cancer.’.

Round 2

Reviewer 2 Report

The revised manuscript has been significantly improved and incorporated suggestions. This manuscript can be consider for publication. 

Author Response

We would like to thank the expert reviewer for critically evaluating our manuscript and providing us with valuable criticism that has helped us to improve the quality of our work. We have thoroughly checked our manuscript on English and typos and changed it where necessary.